# Update in the Treatment of Pleural Tumors: Robotic Surgery Combined with Hyperthermic Intrathoracic Chemotherapy

**DOI:** 10.3390/cancers16091691

**Published:** 2024-04-26

**Authors:** Gaetano Romano, Carmelina Cristina Zirafa, Ilaria Ceccarelli, Gianmarco Elia, Federico Davini, Franca Melfi

**Affiliations:** 1Minimally Invasive and Robotic Thoracic Surgery, Robotic Multispecialty Center of Surgery, University Hospital of Pisa, 56124 Pisa, Italy; c.zirafa@gmail.com (C.C.Z.); i.ceccarelli19@gmail.com (I.C.); davinifederico@gmail.com (F.D.); franca.melfi@unipi.it (F.M.); 2Surgical, Medical, Molecular and Critical Care Pathology Department, University Hospital of Pisa, 56124 Pisa, Italy; gianmarco.elia.7@gmail.com

**Keywords:** pleural recurrences, HITHOC, mesothelioma, thymoma, robotic surgery, thoracic neo-plasms

## Abstract

**Simple Summary:**

The role of surgery in the combined treatment with hyperthermic intrathoracic chemotherapy of thymoma recurrences and pleural mesothelioma has assumed a controversial role in the context of the multimodal treatment of these diseases, especially when minimally invasive techniques are considered. The authors aim to propose future directions in the approach to these pathologies by placing robotic surgery at the center of reflection, starting, however, from a review of the literature available to date.

**Abstract:**

(1) Background. Intracavitary hyperthermic chemotherapy (HITHOC) remains part of the complex mosaic that is the multimodal approach for advanced stage thymoma and pleural malignancies. However, robotic pleurectomy/removal of pleural lesions in combination with intrathoracic chemotherapy is not currently being investigated. The aim of this study is to evaluate the safety of robotic pleurectomy/removal of relapses and HITHOC in patients with pleural recurrence of thymoma or MPM. (2) Methods: The data of nine consecutive patients affected by thymoma relapses or MPM who underwent robotic surgery in combination with HITHOC from February 2017 to November 2022 were collected and analyzed. Surgery performed prior to intrathoracic infusion of high-temperature chemotherapy consisted of removal of recurrences (three patients) or pleurectomy (six patients). All surgeries were performed with a four-port, fully robotic technique. (3) Results: No intraoperative complications occurred. No renal complications related to infusion were recorded. One patient, who underwent pleurectomy for MPM, had a grade II Clavien–Dindo postoperative complication. Oncological follow-up showed results in line with the literature. (4) Conclusions: With the limitation of the small number of patients, robotic surgery in combination with HITHOC seems to be safe in patients with pleural relapses of thymoma and early-stage MPM.

## 1. Introduction

### 1.1. Pleural Mesothelioma

Malignant Pleural Mesothelioma (MPM) is an uncommon malignant neoplasm that arises from the cells of the pleural serosa. Due to its aggressiveness, the therapeutic strategies of mesothelioma have always been very limited, and the prognosis is still considered to be poor [1]. According to the World Health Organization, occupational or para-occupational asbestos exposure is recognized as the leading cause related to mesothelioma [2]. In fact, WHO data from 2020 confirms that 107,000 workers worldwide have died from asbestos exposure and that 125 million people have been exposed to asbestos fibers at least once. Due to the long latency period in pathology expression after exposure, the incidence of malignant mesothelioma has been recorded in a continuous increase in the last decade, with a peak in 2020 [3]. The prognosis of MPM is dramatically poor and ranges from 8 to 14 months after diagnosis, with a more favorable outcome in women than in men [4]. Several studies have demonstrated the correlation between histological subtype and prognosis: the epithelioid form is associated with better survival (14.4 months) when compared to sarcomatous or biphasic mesothelioma (5.4 months) [5].

The treatment of MPM has always been widely debated. Due to its histological features, its aggressiveness, and the average age at diagnosis, no single therapeutic pathway is available to date. Nevertheless, MPM can be the subject of multimodal therapies, aimed at local control of the disease and symptoms, which significantly affect the patient’s quality of life. Furthermore, a non-negligible number of novel therapies are entering clinical practice, opening the scenario for more effective therapeutic options in the future. In fact, in 2021, Tsao and colleagues presented the SWOG1619 (S1619) trial, which aims to demonstrate the feasibility of the association of neoadjuvant chemo-immunotherapy with cisplatin-pemetrexed and atezolizumab in resectable patients diagnosed with epithelioid or bi-phasic mesothelioma. The results are encouraging at present, but for a confirmation of the reproducibility, more prolonged follow-up periods, for the evaluation of the oncologic outcomes and the toxicity, will be necessary [6]. In addition, a recent randomized controlled comparative trial showed impressive overall survival results with immunotherapy (nivolumab plus ipilimumab) compared to chemotherapy (platinum plus pemetrexed) in unresectable MPM, as well as comparable outcomes regarding adverse events and com-plications [7]. The use of surgery is still debated in the mosaic of multimodal treatment of mesothelioma. The objective of a surgical approach to the patient affected by MPM is the macroscopic excision of the disease, therefore considering that complete microscopic eradication is not an achievable goal. Surgery in MPM assumes a transversal role ranging from the histological diagnosis to the curative intent, up to palliation. The NCCN guidelines indeed recognize the role of surgery in pleural biopsy for the diagnosis of MPM, as well as macroscopic removal of the tumor by pleurectomy/decortication (PD) or extrapleural pneumonectomy (EPP) [8]. Therefore, the orientation of clinicians has increasingly been directed towards less invasive and debilitating therapies, to reduce the deterioration of the clinical condition and the quality of life of the patients, without impacting oncological outcomes. For all these reasons, the use of high-temperature chemotherapy in conjunction with cytoreductive surgery, first in the abdomen, and then in the thorax, has found widespread use since the 1980s. In fact, it has been amply demonstrated that during intraoperative chemo-hyperthermia (HITHOC), the high temperature (40–43° Celsius) can improve the permeability of cell membranes, favoring the locally cytotoxic action of the chemotherapy as proven by Schaff and colleagues in an in vitro study [9]. Furthermore, the possibility of local chemotherapy avoids the potentially harmful effects of systemic therapies. The first experience of the use of intracavitary cyto-toxic agents in association with elevated temperature dates to the 1980s, for the treatment of a pseudomyxoma of the peritoneum [10]. Additionally, in the treatment of thoracic neoplasms, the first experience of the use of intracavitary chemotherapy for MPM is reported in the study by Rusch in 1992 [11]. Subsequently, in 1999, Ratto and colleagues demonstrated largely positive pharmacokinetics and few systemic effects on the organism in 10 cases of stage I-II MPM treated with hyperthermic intrathoracic perfusion using cisplatin [12]. Based on the positive results of the previously cited studies, Jarvinen and colleagues published a systematic review on this topic in 2021, analyzing a total of 11 observational studies which focused on the comparison between MPM patients who underwent surgery followed by HITHOC and control patients who were not subjected to HITHOC. The review revealed a statistical significance in terms of survival, favoring the HITHOC group, particularly in patients affected by epithelioid mesothelioma [13].

In 2021, Migliore and colleagues analyzed literature data from 2002–2019 on patients who underwent cytoreductive surgery and HIT-HOC in mesothelioma. The safety and feasibility of the technique, together with the positive outcomes, underlined the im-portance of the adjuvant role of HIT-HOC and support its inclusion in international guide- lines [14]. Two years later, the same authors published the results of a pilot study comparing thoracoscopic pleurectomy and decortication (P/D) with chemical pleurodesis alone, demonstrating a superior survival rate in the VATS P/D plus HIT-HOC group [15].

Therefore, taking into account the literature data, cytoreductive surgery with HITHOC is routinely considered in the clinical practice of several centers.

The HIT-HOC protocols used for mesothelioma in other centers are summarized in Table 1.

### 1.2. Thymoma and Thymic Carcinoma

Thymoma is a rare malignant neoplasm that represents a large part of anterior mediastinal tumors (47%) [16]. Thymoma originates from thymic epithelial cells, and it is often associated with several immunologic disorders such as myasthenia gravis, red cell aplasia and connective tissue diseases. Particularly, about 30% of patients with thymoma are affected by myasthenia gravis and this correlation allows an early diagnosis of thymic diseases. The incidence of thymoma is between 0.13 and 0.32/100.000/year and the most affected age group is the middle age, 45–55 years [17,18].

The strategy of treatment of thymoma is based on the possibility of radical resection and surgery represents the gold standard in early stages, according to the 9th edition of TNM classification for thymic tumors [19]. On the other hand, the treatment of advanced stages and the treatment of TC is still strongly debated, making the role of surgery integrated into a multimodal approach. In this regard, Modh and colleagues investigated patients affected by stage III–IVa Masaoka–Koga thymoma and patients affected by thymic carcinoma in a retrospective study published in 2016. The authors made a comparison between patients treated with a three-modality treatment (surgery, chemotherapy, and radiotherapy) and patients treated with a no-three modality protocol. The chemotherapy was performed pre- and/or post-operatively and the most common regimen was cyclophosphamide, doxorubicin, and cisplatin. This study shows the greatest benefit from three-modality therapy, including improved OS in patients with stage III disease, although no statistically significant differences were found according to the aggressiveness of treatment received [20]. Multimodal treatment protocols, including systemic medical therapy in combination with surgery, have improved the therapeutic options not only in advanced thymic tumors but also in recurrences. Pleural dissemination is the most common localization of relapses (75%) after the first surgical treatment [21] and the resection of the pleural recurrence is a major predictor of favorable outcomes in this setting, with the possible integration of systemic therapy. Despite the absence of many randomized trials, HITHOC is indicated for patients with stage IVa thymoma, characterized by pleural or pericardial dissemination [22], in which the chemo-hyperthermia (42 °C) is performed after the excision of the relapses. In 2001, Refaely and colleagues conducted the first study focused on the role of surgical resection and intrathoracic perfusion of chemotherapy for stage IVa thymic malignancies, describing positive results in terms of locoregional disease control and postoperative morbidity rate [23]. Similar outcomes were reported by De Bree et al., one year later. In both of these studies, no mortality rate was reported; concerning the surgical complications, all the authors described a low rate of events and particularly they reported bleeding, fever, and air leak without hemodynamic or respiratory events during the procedure [24]. In 2015, Ambrogi published a prospective study of a single-center experience of patients with pleural recurrence of thymoma who underwent surgery followed by HITHOC. This study demonstrated that HITHOC was feasible in all cases, with a postoperative morbidity rate of 38%. Furthermore, 85% of patients were alive after a mean follow-up period of 64.6 months; this result is in line with the literature, reporting a five-year survival rate after surgery, between 30 and 75% in different series, with or without neoadjuvant chemotherapy [25]. More recently, in 2023, a systematic review was published by Vandaele et al. focused on the role of cytoreductive surgery combined with HITHOC for pleural disseminated thymoma (TPR) or de novo Masaoka stage IV thymoma (DNT). According to this review, the Disease-Free Interval (DFI) ranged from 6 to at least 88 months in the whole study population cohort, and a five-year survival rate ranged from 70% to 92% [26]. In addition, a review by the Mayo Clinic group in 2023 highlighted the importance of hyperthermic intrathoracic chemotherapy in the local control of pleural pathology, including thymoma recurrence, and emphasized the need for standardization of both indications and techniques. The study led to the establishment of an international task force to promote uniformity in the use of HITHOC for optimal patient outcomes [27]. Despite the fact that multicentric randomized trials are still lacking, the previously published studies seem to support the role of surgery combined with HITHOC in this stage of thymic disease. Therefore, this combined approach could be considered a promising treatment with the potential to improve disease control without compromising the postoperative outcomes [26].

To date, the studies concerning the role of surgery and HITHOC for the pleural dissemination of thymoma or mesothelioma were predominantly conducted on patients treated with the open technique. Minimally invasive surgery could have a positive impact on the outcomes of patients eligible for pleurectomy/removal of pleural lesions plus HITHOC, minimizing the surgical trauma, with a lower complication rate and faster recovery, reducing the risk of delays in starting adjuvant therapy. Robotic surgery is currently the most advanced form of minimally invasive surgery, allowing complex procedures to be performed with high precision. Thanks to its features, robotic surgery also allows access to remote areas, with a complete and accurate exploration of the pleura. The focus of this study is to analyze a single-center experience in the robotic treatment of mesothelioma and pleural thymoma relapses combined with HITHOC.

The HITHOC protocols used for tymoma relapses in other centers are summarized in Table 2.

The primary resource used to select papers for citation in this article was the PubMedNCBI database. The search strategy included the use of specific keywords, including “HITHOC”, “intrathoracic chemo-hyperthermia”, “thymoma recurrence”, “mesothelioma”, “pleurectomy” and “minimally invasive surgery”. To refine the search results, these keywords were combined using the conjunction “AND”. Wherever possible, priority was given to articles published within the last 5 years. This approach aimed to include the most recent research and advances in the field, ensuring the inclusion of up-to-date information related to HITHOC and minimally invasive surgical procedures for thymoma recurrence and mesothelioma.

## 2. Robotic Surgery Combined with HITHOC, Our Experience

### Materials and Methods

The data of patients who underwent robotic surgery in combination with HITHOC, from 2017 to 2022, were analyzed. Patients affected by clinical stage I MPM and with pleural recurrence of thymoma (clinical stage IVa) were selected. Clinical characteristics of the patients, surgical, postoperative, and oncological results were collected. The surgical parameters analyzed were time of surgery, rate of conversion, post-operative complications (according to Clavien–Dindo classification), chest tube duration, and the length of hospital stay.

All patients were studied preoperatively with a total body CT (computed tomography) scan and PET (positron emission tomography). All the cases were evaluated collegially by the tumor board. None of the patients with thymoma underwent preoperative biopsy.

The oncological results are expressed in terms of adjuvant treatments with the number of therapy cycles, the rate and the date of eventual relapses, the treatment of the relapse and the overall survival.

The surgery was performed with a four-access robotic technique under general anesthesia and selective orotracheal intubation. The chemotherapy drugs used during the surgery are 80 mg of Cisplatin and 25 mg of Epirubicin, based on the patient’s body surface area, expressed in square meters, for 60 min at a constant target temperature of 42 °C. The dedicated circuit for the continuous infusion of chemotherapy, diluted in crystalloids, is connected directly to the two pleural drains positioned at the end of surgery, through two of the surgical accesses. A 0.4 cm diameter thermometric probe is introduced into the pleural cavity to constantly monitor the temperature. The perfusion of the chemotherapy drugs, cisplatin (80 mg/m^2^) and epirubicin (25 mg/m^2^), was performed at the end of the cytoreductive phase for 60 min at a constant temperature of 42 °C. The therapeutic peri-operative protocol presented in this experience consists of 2000 cc of hydration with different electrolytic solutions, pantoprazole, loop diuretic, and 8 mg of corticosteroid the day before surgery. Subsequently, the day of surgery, 3000 cc of hydration, pantoprazole, loop diuretic, and corticosteroid were administered. At last, from the first to the 5th postoperative day, 2000 cc of hydration with gastric protection, loop diuretic, and 4 mg of corticosteroid were infused. In addition, the patients are treated with enoxaparin during the postoperative stay. The detailed hydration protocol is shown in Table 3.

The surgical technique of robotic pleurectomy with HITHOC is shown in the short video, while Figure 1 shows the removal of a pleural thymoma recurrence.

## 3. Results

The data of nine patients affected by thymoma relapses or MPM who underwent robotic surgery in combination with HITHOC, from February 2017 to November 2022, were collected and analyzed.

No intraoperative bleeding or renal complications related to drug infusion were observed.

Lung–diaphragm–pericardium-sparing pleurectomy was performed in six patients (4MPM, two thymoma recurrences) following previous experience reported in the literature [28], while three patients with thymoma recurrence underwent removal of pleural lesions. The mean operative time in the pleurectomy group (including docking time and perfusion) was 320 min (range 230–455) and 231.7 min (range 175–315) in the thymoma patients.

The average amount of fluid drained in the first 24 h after HITHOC for both MPM and thymoma relapses was 315 milliliters (mL). The amount drained between 24 and 48 h after the procedure was 290 mL.

Looking at the groups separately, the average amount of fluid drained in the first 24 h was 430 mL in patients with MPM and 220 mL in patients with thymoma recurrence. Additionally, the amount of fluid drained between 24 and 48 h was 5000 mL for MPM and 1400 mL for thymoma recurrence.

Epithelioid mesothelioma was diagnosed in four (44%) patients; three were females (75%) and one was male (25%), with a mean age of 64.5 (SD 9, range 55–78). One conversion (25%) to posterior-lateral thoracotomy was reported, due to the obliteration of the pleural space.

In all cases, the drugs used for infusion were Cisplatin and Epirubicin.

After intraoperative frozen section analysis, one patient (25%) with MPM underwent surgery with HITHOC. In three (75%) patients with MPM, thoracoscopic diagnosis was performed preoperatively. In the MPM group, a postoperative complication was observed in two (50%) cases, represented by one (25%) case of Atrial Fibrillation, Grade II according to Clavien–Dindo classification, treated with cardioversion and one (25%) case of prolonged air leak (Grade I). The mean duration of the chest tube was six days (SD 1.4) (range 5–8); the mean length of stay was nine (SD 1.7) (range 7–11). The adjuvant therapy was administered in three (75%) patients affected by MPM; all cases consisted of six cycles of Pemetrexed and Cisplatin. During the follow-up, two (50%) patients showed a relapse, 36 and 20 months after the procedure, respectively. Both recurrences involved the ipsilateral pleura and one of them also involved the ipsilateral lung; the treatment of the relapse was a combination of radio-therapy and Pemetrexed. After a mean follow-up of 52 months (SD 25.2) (range 15–72), all the patients were alive.

The patients with the diagnosis of cIVa thymoma treated with HITHOC were five (56%); three men (60%) and two women (40%), with a mean age of 50 (SD 13.2, range 34–62). Previously, three patients (60%) underwent thymectomy by sternotomy, while in two cases (40%), the procedure was performed by a robotic approach. The thymoma histology was represented by B1 in one case (20%), B2 in two patients (40%), B3 in one case (20%) and thymic carcinoma in one case (20%). Four patients (80%) were treated with an adjuvant protocol after the thymectomy. The surgical procedure combined with HITHOC was pleurectomy in two cases (40%) and removal of pleural recurrences occurred in the other three cases (60%). No intraoperative complications were registered. Concerning postoperative complications, one patient (20%) underwent relapse removal and presented prolonged air leak (Grade I). The time of chest tube duration resulted in six days (SD 2.2, range 4–10); the mean length of stay was six days (SD 0.8, range 6–8). After the surgical procedure, one (20%) patient was treated with chemotherapy plus radio-therapy because of a mediastinal residual disease, one was treated (20%) with adjuvant chemotherapy. During the follow-up, three (60%) patients showed an ipsilateral pleural recurrence, and one (20%) had a pleural and lung contralateral relapse. The median time of relapse was 23 months (DS 17.3). One (20%) of them were treated with chemotherapy while three (60%) patients underwent a second surgical procedure with HITHOC, in two cases on the same side and one case on the opposite side. After a mean follow-up of 58 months (SD 6.8, range 48–69), all the patients are alive.

## 4. Discussion

The advent of robotic surgery has been a game-changing improvement compared to the minimally invasive approaches of the past [29].

Over the years, in the thoracic field, the robotic approach has allowed us to overcome all the limits of video-assisted thoracoscopic surgery, thanks to the 3D high-definition vision, the magnification of the image up to 10 times, the seven degrees of freedom of the instruments, and the filtration of physiologic hand tremors.

The robotic system has become progressively widespread, and published studies have demonstrated improved intraoperative and postoperative outcomes and extension of surgical indications [30]. Furthermore, robotic surgery can play a role in patients affected by early stage of pleural mesothelioma or in cases of pleural recurrence of thymoma. In fact, in the scientific literature, recurrence with a pleural dissemination is reported also after radical resection of thymoma [31].

As Ruffini et al. described in their article published in 1997, the total resection of the recurrences offers the best chance of long-term survival, when compared to the medical treatment. Moreover, the authors underline how the surgical removal of the relapses could be difficult due to the second surgery and the disseminated localization of the pleural disease, which may be undetectable at the preoperative CT scan [32]. The technical characteristics of robotic systems can help to overcome these challenges, as Cavaliere reported in a case report in 2017. The robotic technique was found to be useful and effective in exploring the whole chest cavity with the aim of also identifying small and previously undetected pleural relapses [33]. Agustin and coworkers reported the same advantages in the use of robotic technique in a paper published in 2006: the technical features of the da Vinci™ system are most advantageous in tiny and difficult-to-reach anatomical regions with a low rate of intraoperative complications [34]. Concerning the MPM, the first case of robotic pleurectomy in patients with MPM was described in a case report published in 2015. The author observed a higher freedom of motion of the instruments in the chest cavity and low blood loss during the procedure, confirming how this approach could be considered feasible and safe even for demolitive surgeries [35]. According to Optiz and colleagues, surgery for MPM is still considered to be associated with too high morbidity and mortality. The best long-term results are achievable just with multimodal treatment, and surgery has its pivotal role [36]. In our experience, both for patients affected by early-stage pleural mesothelioma or stage IVa thymoma, no intraoperative complications were reported in association with a low rate of postoperative complication, compared to the data in the literature [37]. Moreover, the duration of the chest tube and the length of stay in our patients affected by MPM and treated with pleurectomy and HITHOC with the robotic approach seems to be shorter than the same procedure with other approaches [38,39]. The surgical outcomes achieved by the robotic technique in our case series could represent a fundamental factor for postoperative management: a short hospital stay allows for a faster recovery and an early start for adjuvant therapy. The importance of adjuvant therapy in MPM patients is known from the first publication of Butchart in the early 1980s. As the author described, the barrier to the effectiveness of the adjuvant treatment was the high rate of perioperative morbidity and mortality [40]. Furthermore, as Cao and colleagues discussed in their systematic review, the improvements in surgical techniques and perioperative care have resulted in significantly superior outcomes in patients treated with multi-modality therapy [41].

## 5. Conclusions

In conclusion, the current results highlight the potential efficacy of robotic surgery combined with hyperthermic intraoperative chemotherapy, while acknowledging the limitations of a small patient sample and a relatively short follow-up period for patients with thymoma. The favorable surgical outcomes observed hold promise for integration into a comprehensive multimodal treatment strategy. However, it is important to emphasize the need for additional data to further substantiate these findings and establish a more complete understanding of the long-term benefits and potential challenges associated with this approach in patients with early-stage MPM and thymoma recurrences. Future research will be needed to confirm the results of this experience.

## Figures and Tables

**Figure 1 cancers-16-01691-f001:**
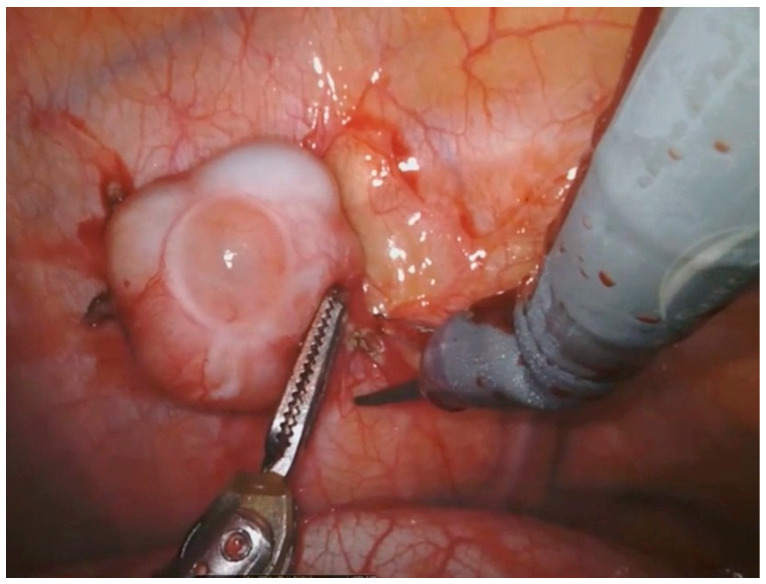
Removal of a pleural thymoma relapse with robotic technique.

**Table 1 cancers-16-01691-t001:** HITHOC protocols used for patients affected by pleural mesothelioma.

Author (Year)	Drugs (Dosage mg/m^2^)	Target Temperature (°C)	Perfusion Time (Min)	Number of Pleural Catheters
Rusch (1992) [11]	Cisplatin (100) Mitomycin (8)	41.5–42	60	3 (2 inflow, 1 outflow)
Ratto (1999) [12]	Cisplatin (100)	41.5	60	2 (1 inflow, 1 outflow)
	Cisplatin (80) Adriamycin (25)	40–42	90	
Jarvinen (2021) [13]	Cisplatin (225)	42	60	2 (1 inflow, 1 outflow)
	Cisplatin (175–225)	42	60	
	Cisplatin (175–225) Gem- citabine (900)	42	120	
Migliore (2023) [15]	Cisplatin (120)	42.5	60	2 (1 inflow, 1 outflow)

**Table 2 cancers-16-01691-t002:** HITHOC protocols used for patients affected by thymoma and thymic carcinoma.

Author (Year)	Drugs (Dosage mg/m^2^)	Target Temperature (°C)	Perfusion Time (Min)	Number of Pleural Catheters
Refaely (2001) [23]	Cisplatin (100–200)	40.3–45	60	2 (1 inflow, 1 outflow)
De Bree (2002) [24]	Cisplatin (50–80) Adriamycin (15–25)	40–41	90	4 (1 inflow, 3 outflow)
Ambrogi (2016) [25]	Cisplatin (80) Doxorubicin (25)	42.5	60	2 (1 inflow, 1 outflow)
Campany (2023) [27]	Cisplatin (150–225)	42	60	4 (2 inflow, 2 outflow)

**Table 3 cancers-16-01691-t003:** Hydration protocol for patients underwent HITHOC.

Day before Surgery	Operative Day	1–5 Post-Operative Day
500 cc of saline solution plus 40 mg of Pantoprazole plus 8 mg of Dexamethasone 1000 cc of Ringer’s Lactate solution 500 cc of saline solution plus 20 mg of Furosemide	500 cc of saline solution plus 40 mg of Pantoprazole plus 16 mg of Dexamethasone 500 cc of saline solution plus 20 mEq/L of Magnesium Sulphate (MgSO_4_) 500 cc of saline solution plus 20 mEq/L of Potassium Chloride (KCl) 500 cc of saline solution plus 20 mg of Furosemide	500 cc of saline solution plus 40 mg of Pantoprazole plus 4 mg of Dexamethasone 1000 cc of Ringer’s Lactate solution 500 cc of saline solution plus 20 mg of Furosemide

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
