# Peer review of "Update in the Treatment of Pleural Tumors: Robotic Surgery Combined with Hyperthermic Intrathoracic Chemotherapy"

_cancers, 2024, doi:10.3390/cancers16091691_

Round 1

Reviewer 1 Report

Comments and Suggestions for Authors

Although the article could have been interesting, it has too many weak points.

1)The introduction is too long and tedious. It should have been restricted to HITHOC and mesothelioma or type IV thymoma to show readers and reviewers why the research topic is worth reading. 

2) Line 242 The therapeutic peri-operative protocol presented in this experience consists of 2000 cc of hydration with different electrolytic solutions 

COMMENT. what does it mean different electrolytic solutions. Authors need to be precise. Which solutions have been used. 

3) Description of the surgical technique is poor. More details are necessary. How many patients underwent P/D and how many eP/D.  Any intraoperative bleeding? How much is the dose of chemiotherapy during HITHOC? Any renal complications? 

4) Nine patients in 6 years is a very low number to say that the procedure has efficacy and safety. Title should therefore be changed.

5) There is no data at all on preoperative work up. PET, Lung function test, preoperative biopsy etc.   WAs biopsy performed via VATS or CT guided? 

6) Although the chest drain was removed at the 6th post op day, why mesothelioma patients spent 3 more days into the hospital while thymoma patients have been sent home the same day? Can you comment?

7) References are incomplete as recent systematic reviews have been missed; furthermore recent original papers with minimally invasive surgery and HITHOC have not been reported. 

8) References have not been written according to journal style

9) It is a speculation to write (line 35-36)  that "considering the results, robotic surgery may represent 35 the surgical approach of choice, in combination with HITHOC, in patients with pleural relapses of 36 thymoma and early-stage MPM". There are no comparative data supporting this personal opinion/conclusion. 

10) This article is not a narrative review and it is not a surgical technique as it lacks of many details. 

11) How the papers included in the manuscript have been chosen? 

12) there are no intraoperative pictures or a short video which could attract the readers. 

Thank you for sending this paper to cancer

Comments on the Quality of English Language

There are many typos. 

Author Response

Dear colleague,

we express our gratitude for your meticulous review of our manuscript intended for "Cancers". We appreciate your consideration of our work for inclusion in this esteemed journal. In response to your insightful comments, we present the following clarifications and revisions.

Although the article could have been interesting, it has too many weak points. 

1)The introduction is too long and tedious. It should have been restricted to HITHOC and mesothelioma or type IV thymoma to show readers and reviewers why the research topic is worth reading. 

Thank you for your suggestion. We have revised the introduction by eliminating several sections deemed unrelated to the article focus, streamlining and synthesizing aspects that we consider pertinent for reader comprehension.

2) Line 242 The therapeutic peri-operative protocol presented in this experience consists of 2000 cc of hydration with different electrolytic solutions 

COMMENT. what does it mean different electrolytic solutions. Authors need to be precise. Which solutions have been used. 

Thank you for your suggestions, we have incorporated tables that summarize the hydration protocol employed at our center. For completeness we explain that the protocol consists of:

Day before surgey:

  • 500 cc of Saline Solution plus 40 mg of Pantoprazole plus 8 mg of Dexamethasone;
  • 1000 cc of Ringer’s Lactate solution;
  • 500 cc of Saline Solution plus 20 mg of Furosemide.

Operative day:

  • 500 cc of Saline Solution plus 40 mg of Pantoprazole plus 16 mg of Dexamethasone;
  • 500 cc of Saline Solution plus 20 mEq/L of Magnesium Sulphate (MgSO4);
  • 500 cc of Saline Solution plus 20 mEq/L of Potassium Chloride (KCl);
  • 500 cc of Saline Solution plus 20 mg of Furosemide.

For five days after surgery, the patient is hydrated according to the protocol with:

  • 500 cc of Saline Solution plus 40 mg of Pantoprazole plus 4 mg of Dexamethasone;
  • 1000 cc of Ringer’s Lactate solution;
  • 500 cc of Saline Solution plus 20 mg of Furosemide.

  • Description of the surgical technique is poor. More details are necessary. How many patients underwent P/D and how many eP/D.  Any intraoperative bleeding? How much is the dose of chemiotherapy during HITHOC? Any renal complications? 

All patients (100%) affected by MPM underwent lung-diaphragm-pericardium-sparing pleurectomy [1] . Two (40%) patients affected by thymoma relapses were treated with pleurectomy while the other 3 (60%) underwent excision of the relapses. No intraoperative bleedings were registered.  The chemotherapy drugs used during the surgery are: 80 mg of Cisplatin and 25 mg of Epirubicin, dosed based on the patient's body surface area, expressed in square meter, for 60 minutes at a constant target temperature of 42°C. No renal complications were registered. The mentioned considerations are included in the Methods and Results sections.

1 Ambrogi MC, Bertoglio P, Aprile V, Chella A, Korasidis S, Fontanini G, Fanucchi O, Lucchi M, Mussi A. Diaphragm and lung-preserving surgery with hyperthermic chemotherapy for malignant pleural mesothelioma: A 10-year experience. J Thorac Cardiovasc Surg. 2018 Apr;155(4):1857-1866.e2. doi: 10.1016/j.jtcvs.2017.10.070. Epub 2017 Nov 1.

4) Nine patients in 6 years is a very low number to say that the procedure has efficacy and safety. Title should therefore be changed.

Thanks for your suggestion, the title have been changed.

  • There is no data at all on preoperative work up. PET, Lung function test, preoperative biopsy etc.   WAs biopsy performed via VATS or CT guided? 

All patients were studied preoperatively with a total body CT scan and PET-CT. All the cases were evaluated collegially by the tumor board. No patients affected by Thymoma underwent a preoperative biopsy.

1 (25%) patient with MPM underwent pleurectomy with HITHOC after an intraoperative frozen section examination. In 3 (75%) patients affected by MPM the preoperative diagnosis was done by VATS.

These considerations are mentioned in the methods section.

  • Although the chest drain was removed at the 6th post op day, why mesothelioma patients spent 3 more days into the hospital while thymoma patients have been sent home the same day? Can you comment?

One (25%) patient with MPM experienced a postoperative complication—atrial fibrillation—necessitating extended observation and cardiological examinations. Another patient (25%) in this group underwent a conversion from the robotic technique to open surgery, which accounted for the prolonged stay. In this instance, the patient required additional days to achieve complete autonomy and effective pain control. In the last two cases of mesothelioma patients, the extended stay can be attributed to the general condition of individuals with this type of disease. As mentioned in the paper, patients with MPM are generally more fragile and typically older than those experiencing thymoma relapses.

  • References are incomplete as recent systematic reviews have been missed; furthermore recent original papers with minimally invasive surgery and HITHOC have not been reported. 

Thank you for your valuable suggestion, we have enriched the reference concerning MPM and HITHOC, mentioning the following studies:

  1. Migliore M, Ried M, Molins L, Lucchi M, Ambrogi M, Molnar TF, Hofmann HS. Hyperthermic intrathoracic chemotherapy (HITHOC) should be included in the guidelines for malignant pleural mesothelioma. Ann Transl Med. 2021 Jun;9(11):960. doi: 10.21037/atm-20-7247.
  2. Migliore M, Fiore M, Filippini T, Tumino R, Sabbioni M, Spatola C, Polosa R, Vigneri P, Nardini M, Castorina S, Basile F, Ferrante M; University of Catania Study Group of Malignant Pleural Mesotelioma. Comparison of video-assisted pleurectomy/decortication surgery plus hyperthermic intrathoracic chemotherapy with VATS talc pleurodesis for the treatment of malignant pleural mesothelioma: A pilot study. Heliyon. 2023 May 25;9(6):e16685. doi:
  3. Campany ME, Reck dos Santos PA, Donato BB, Alwardt CM, Ernani V, D’Cunha J, Beamer SE. Hyperthermic intrapleural chemotherapy: an update. J Thorac Dis 2023;15(9):5064-5073. doi: 10.21037/jtd-23-454

  • References have not been written according to journal style

The references have been corrected following the style present in other papers published in the journal. We welcome any additional suggestions

  • It is a speculation to write (line 35-36)  that "considering the results, robotic surgery may represent 35 the surgical approach of choice, in combination with HITHOC, in patients with pleural relapses of 36 thymoma and early-stage MPM". There are no comparative data supporting this personal opinion/conclusion. 

In agreement with the comments of the other reviewers, the sentence has been modified as follows:

“With the limitation of the small number of patients, robotic surgery in combination with HITHOC seems to be effective in patients with pleural relapses of thymoma and early-stage MPM”

  • This article is not a narrative review and it is not a surgical technique as it lacks of many details. 

In agreement with your comments, we have eliminated the definition "narrative review" by setting the article differently and providing more details about the surgical approach used in our experience.

  • How the papers included in the manuscript have been chosen? 

In the process of selecting papers for citation in this article, the PubMed NCBI database was employed as the primary resource. The search strategy involved the use of specific keywords, including "HITHOC," "intrathoracic chemo-hyperthermia", "thymoma relapses", "mesothelioma",“pleurectomy” and "minimally invasive surgery". To refine the search results, these keywords were combined using the conjunction "AND."Emphasis was placed on prioritizing articles published within the last 5 years, whenever possible. This approach aimed to incorporate recent research findings and advancements in the field, ensuring the inclusion of up-to-date information related to HITHOC, and minimally invasive surgical procedures for thymoma relapses and mesothelioma.

For completeness, we have included these remarks in the Introduction section.

  • there are no intraoperative pictures or a short video which could attract the readers. 

A short video of a mesothelioma patient and a photo of a thymoma relapse removal is now attached to the paper.

Reviewer 2 Report

Comments and Suggestions for Authors

Thanks to the authors for giving me the opportunity to review their manuscript. It is very well written, very complete

I have a few comments:

You write in the abstract : “robotic surgery may represent the surgical approach of choice,”

I think the conclusion of the article is more appropriate « robotic surgery combined with hyperthermic intraoperative chemotherapy seems to be effective”

Why didn't you use sodium thiosulfate aiming to reduce the incidence and severity of cisplatin-related nephrotoxic effects?

Can you specify the average amount of drainage ?

Comments on the Quality of English Language

/

Author Response

Dear Colleague,

We wanted to express our sincere gratitude for your invaluable assistance in reviewing our article for publication in Cancers. Your willingness to lend your expertise and time for the revision process is truly appreciated.

Below we report the answers to your suggestions.

Thanks to the authors for giving me the opportunity to review their manuscript. It is very well written, very complete

I have a few comments:

You write in the abstract : “robotic surgery may represent the surgical approach of choice,”

I think the conclusion of the article is more appropriate « robotic surgery combined with hyperthermic intraoperative chemotherapy seems to be effective”

Thank you for your comment, we have corrected the sentence with your valuable suggestion.

Why didn't you use sodium thiosulfate aiming to reduce the incidence and severity of cisplatin-related nephrotoxic effects?

Limited to the small number of our cases, no side effects related to nephrotoxicity from chemotherapy have occurred and our protocol does not include the use of thiosulfate, however, we will take your valuable suggestion into account in the future.

Can you specify the average amount of drainage?

The average amount of fluid drained in the first 24 hours after HITHOC, for both MPM and thymoma relapses, was 315 millimeters (ml). The amount drained 24 to 48 hours after the procedure was 290 ml.

Concerning the groups separately: the average amount of fluid drained in the first 24 hours resulted in 430 ml in patients with MPM and in 220 ml in patients affected by thymoma relapses. In addition, the amount drained 24 to 48 hours after surgery resulted in 500 ml in MPM and in 140 ml in patients with thymoma relapses.
We have added these considerations in results section.

Reviewer 3 Report

Comments and Suggestions for Authors

The authors present a narrative review and their own experience with hyperthermic intrathoracic chemotherapy  (HITHOC) for pleural tumours, more specifically pleural mesothelioma and thymic epithelial tumours with pleural dissemination. The study question is relevant to current practice as the role of HITHOC has not precisely determined and is still under investigation.

Comments:

- line 23: the authors state they present a narrative review but no specific guidelines are followed. These should be further elaborated. An example include:

https://towson.libguides.com/expert-reviews/narrative-literature-reviews

- Cancers is a specialist journal; so, their long introduction on pleural mesothelioma and thymic tumors is not necessary as these large parts (lines 43-125 and 150-185) resemble more a chapter of a medical oncology text book for students

- instead, I propose the authors mainly focus on recent data of HITHOC and provide some level of evidence how to incorporate this technique within the multimodality treatment of these pleural tumours, not only in robotic surgery

- the manuscript only consists of text: some tables on HITHOC studies separating pleural mesothelioma and thymic tumours, and figures of their HITHOC technique which is well described, should be provided as they are of special interest to the readers

- regarding mesothelioma, the recent randomised controlled CM 743 trial by P. Baas et al. Lancet 2021; 397 (10272): 375-386 evaluating double immunotherapy which has become standard treatment for unresectable pleural mesothelioma, should be mentioned

- also, reference to 9th edition of TNM classification for thymic tumours which should preferentially be used, should be inserted

Comments on the Quality of English Language

Minor spellingn errors only, as e.g. line 122: "option" instead of "options"

Author Response

Dear colleague,

We wanted to extend our sincere appreciation for your generous contribution in reviewing our article for potential inclusion in “Cancers”. Your willingness to dedicate your time and expertise to this endeavor has been invaluable.

Below the comments to your precious suggestions are reported.

Reviewer 3

Comments:

- line 23: the authors state they present a narrative review but no specific guidelines are followed. These should be further elaborated. An example include:

https://towson.libguides.com/expert-reviews/narrative-literature-reviews

Thank you for your valuable observation,

in accordance with reviewer Number 1, we have structured our paper differently as an original article

- Cancers is a specialist journal; so, their long introduction on pleural mesothelioma and thymic tumors is not necessary as these large parts (lines 43-125 and 150-185) resemble more a chapter of a medical oncology text book for students

- instead, I propose the authors mainly focus on recent data of HITHOC and provide some level of evidence how to incorporate this technique within the multimodality treatment of these pleural tumours, not only in robotic surgery

In response to the constructive feedback provided by other reviewers and taking into consideration their concerns, we have restructured the introduction of our paper. Our aim was to enhance readability by eliminating certain tedious sections while placing a greater emphasis on the surgical treatment associated with surgery plus HITHOC.

We trust that this revised approach will be well-received, and we remain open to further guidance and suggestions. Your input is valuable, and we appreciate the collaborative effort to refine and improve our work.

- the manuscript only consists of text: some tables on HITHOC studies separating pleural mesothelioma and thymic tumours, and figures of their HITHOC technique which is well described, should be provided as they are of special interest to the readers

Thanks for the valuable suggestion. In order to enrich the article and arouse greater interest in the reader, three summary tables have been included about the articles cited and the protocol available in our center. In addition, a short video and a photo concerning the surgical technique are now available.

- regarding mesothelioma, the recent randomised controlled CM 743 trial by P. Baas et al. Lancet 2021; 397 (10272): 375-386 evaluating double immunotherapy which has become standard treatment for unresectable pleural mesothelioma, should be mentioned

Thanks for your valuable comment, we added the article in the introduction section.

- also, reference to 9th edition of TNM classification for thymic tumours which should preferentially be used, should be inserted

We are sorry for the inattention. The 9th edition of TNM classification is now cited in the article.

Round 2

Reviewer 1 Report

Comments and Suggestions for Authors

Dear Authors

thank you for the extensive review. I have more comments.  

The abstract should be modified as the introduction is too long, while methods and results are incomplete. It need to be rewritten.

Moreover in the abstract you wrote in the conclusion: With the limitation of the small number of patients, robotic surgery in combination with HITHOC seems to be effective in patients with pleural relapses of thymoma and early-stage MPM

COMMENT: this is different from what Authors wrote at the end of the paper. As the authors have no data on long term survival the word "effective" should be changed in safety.

Line 207 .....who underwent robotic surgery in combination with HITHOC, from 2017 to 2022,

COMMENT: authors are invited to add months for the initial and final date of the study 

Lines 217-219 and 224-227 

COMMENT: these lines are result of the study. They should be added in the results chapter and not in the method. 

Line 224-225 All patients (100%) affected by MPM underwent lung-diaphragm-pericardium-sparing pleurectomy, following previous experience reported in the literature. 

COMMENT: please add reference

Line 286. The mean time of surgery, including the docking time and HITHOC has resulted in 269 minutes (SD 70).

COMMENT: it is not correct to include in the same basket patients who underwent total or subtotal  pleurectomy (6 pts) and 3 patients who had only an excision of the relapses. This creates confusion and an alteration of the mean operation duration, hospital stay, complications. 

COMMENT: please revise operative time and complications  in both groups (pleurectomy vs excision of the lesion)

Other 

It is not always possibile to differenciate mesothelioma or lung cancer on a frozen section. Which method have been used? 

Moreover it seems not justify to use the robot for excision of the relapses, it seems a waste of resources. Can you comment?

Thank you for sending this paper in Cancers

Author Response

The abstract should be modified as the introduction is too long, while methods and results are incomplete. It need to be rewritten.

Thank you for your comments, the abstract has been shortened and revised.

Moreover in the abstract you wrote in the conclusion: With the limitation of the small number of patients, robotic surgery in combination with HITHOC seems to be effective in patients with pleural relapses of thymoma and early-stage MPM

COMMENT: this is different from what Authors wrote at the end of the paper. As the authors have no data on long term survival the word "effective" should be changed in safety.

“Effective” has been changed to “safety”.

Line 207 .....who underwent robotic surgery in combination with HITHOC, from 2017 to 2022,

COMMENT: authors are invited to add months for the initial and final date of the study 

The months were inserted both in the abstract and in the body of the text.

Lines 217-219 and 224-227 

COMMENT: these lines are result of the study. They should be added in the results chapter and not in the method. 

It was corrected as you suggested.

Line 224-225 All patients (100%) affected by MPM underwent lung-diaphragm-pericardium-sparing pleurectomy, following previous experience reported in the literature. 

COMMENT: please add reference

Reference added.

Line 286. The mean time of surgery, including the docking time and HITHOC has resulted in 269 minutes (SD 70).

COMMENT: it is not correct to include in the same basket patients who underwent total or subtotal  pleurectomy (6 pts) and 3 patients who had only an excision of the relapses. This creates confusion and an alteration of the mean operation duration, hospital stay, complications. 

COMMENT: please revise operative time and complications  in both groups (pleurectomy vs excision of the lesion)

The operative time and complications were revised according to your suggestion. Regarding postoperative complications, we included the type of surgery when we mentioned the complication itself.

There was no significant difference in the length of stay in the hospital according to the type of surgery performed.

Other 

It is not always possibile to differenciate mesothelioma or lung cancer on a frozen section. Which method have been used? 

The differential diagnosis was supported by the absence of parenchymal lesions and the patient's reported occupational exposure to asbestos. Frozen section plus possible HITHOC was supported by our tumour board.

Moreover it seems not justify to use the robot for excision of the relapses, it seems a waste of resources. Can you comment?

Robot surgery can combine the advantages of traditional open surgery with a minimally invasive approach. In this case, we believe that the robotic platform could play a crucial role in reaching remote areas of the thorax, where the tenfold magnified 3D vision allows the detection of even small recurrences, ensuring a radical operation from an oncological point of view, combined with all the advantages for the patient associated with a minimally invasive access.

In addition, when performed in high-volume specialized centres, the cost of maintaining the platform can be reduced thanks to the large number of cases performed.

Thank you for the thorough review of our article.

Reviewer 3 Report

Comments and Suggestions for Authors

The revised manuscript underwent major improvements and the authors responded well to the comments that were made.

Comments on the Quality of English Language

only minor speling errors

Author Response

Thank you for your valuable suggestions and for taking the time to review our work.

Round 3

Reviewer 1 Report

Comments and Suggestions for Authors

thank you for your review, but there is still something to change/add. The introduction chapter disappeared!! What the authors call introduction chapter is  material and method. Authors need to add the introduction and Method chapters.

Authors did not convinced me that robot surgery should be used for small pleural recurrence lesions. This is just a speculative sentence of the authors as there is no prove that 3D vision allows the detection of small recurrences. Moreover it is noted that  authors did not add the comment in the text as for example a limitation paragraph. 

Minor: change the word safety in safe in the conclusion of the abstract. 

I

Comments on the Quality of English Language

None 

Author Response

Thank you for your comments. Please find our response in the attachment.

Round 4

Reviewer 1 Report

Comments and Suggestions for Authors

I think that the paper can now be accepted.
I did not see the COI  in the pdf. 
Nevertheless for me it is imperative to accept the manuscript  that  an Accurate conflict of interest (COI) statements is presented to avoid to invalidate the study due to the potential risk of bias. Without a self-declared COI statements with robotic company, I am not going to accept the paper. 
If necessary I can ask the autthors to include the COI as last point of the revision.